# The association between material hardship and physical and mental health among older adults: Multi-channel sequence Approach

Oejin Shin[1]*, Eunsun Kwon[2], Seoyeon Ahn[3], Sojung Park[4]

1 School of Social Work at Illinois State University, Normal, Illinois, United States of America, 2 School of Pharmacy and Health Sciences at Fairleigh Dickinson University, Florham Park, New Jersey, United States of America, 3 Centre for Ageing Research & Education at Duke-NUS Medical School, Singapore, Singapore, 4 Brown School of Social Work at Washington University in St. Louis, Saint Louis, Missouri, United States of America

* oshin@ilstu.edu

## Abstract

Older adults often face financial difficulties due to declining physical and cognitive abilities, and rising healthcare costs exacerbate these issues. Despite a general decrease in poverty, the rate among those aged 65 and older has increased. Material hardship is particularly relevant for older adults due to their complex healthcare needs. However, research on long-term material hardship patterns among older adults is limited and often focuses on only a few dimensions of hardship. This study aims to explore the long-term patterns of material hardship and their impact on well-being using data from the Health and Retirement Study. Employing multi-channel sequence analysis, we model various material hardship trajectories and examine their association with mental and physical health. Five distinct patterns of material hardship were identified, with the Multiply burdened group experiencing the most severe hardships, often linked to females, low education, and poverty. The findings highlight the significant negative effects of persistent material hardship on health, emphasizing the need for targeted policies and support programs to address the unique challenges faced by older adults, especially those related to housing and financial stress.

## Material hardship among older adults

Material hardship focuses on the lack of goods and services considered necessary for a decent standard of living, while poverty is more about the lack of resources [1]. The term "material hardship" is often used interchangeably with related phrases such as financial hardship or economic deprivation. Approximately one in five elderly individuals experienced at least one financial hardship, such as food insecurity, skipped meals, medication cutbacks, difficulty paying bills, and dissatisfaction with one's financial situation [2].

Few studies that have focused on material hardships among older adults have highlighted the negative association between material hardship and well-being. Long-term financial strain across the life course has been linked to negative health outcomes in later life, emphasizing the cumulative impact of financial stress on health and well-being. Using retrospective data,

**Data availability statement:** Yes, it is third-party data. All data files are available from the Health and Retirement Study database (https://hrs.isr.umich.edu). The dataset titles are "Longitudinal RAND HRS" and "HRS Core." This dataset is publicly accessible and can be downloaded by anyone through the website. The authors do not have any special access privileges.

**Funding:** The author(s) received no specific funding for this work.

**Competing interests:** NO authors have competing interests.

research has shown that prolonged financial hardship adversely affects later-life health outcomes [3]. Additionally, the association between material hardship—such as difficulty paying bills, financial stress, medication reduction due to cost, and food insecurity—and self-rated health among adults aged 50 and older was explored using the Health and Retirement Study (HRS). Findings revealed a persistent negative relationship between experiencing material hardship and self-assessed health, even after controlling for demographic characteristics, household income, education, and employment status [4]. Further analysis of six waves of HRS data uncovered varied trajectories of food insecurity and medication cutbacks, which were strongly linked to poorer physical health outcomes. Individuals with prior experiences of these hardships were more likely to report worse health over time [5].

Although existing research provides valuable insights into material hardship and well-being among older adults, the majority of the results are based on only measuring specific aspects of material hardship using 1-2 indicators. This approach may risk overlooking information about the extent of material hardship experienced. Since researchers have conceptualized material hardship as multifaceted [1,6], and it is possible that older adults experience different patterns of hardship over time, study to examine how older adults experience material hardship both concurrently and overtime is urgently needed. Conducting an analysis to discern longitudinal patterns is essential for distinguishing between older adults experiencing persistent challenges across multiple domains and those encountering specific issues. It is imperative to recognize the differing needs of these populations based on the duration and severity of their difficulties. In addition, there is a lack of research on mental health among older adults who experienced material hardship. Existing research mainly focused on physical health alone without the consideration of the mental health circumstances among older adults. As understanding the quality of life of older adults encompasses both mental and physical health [7], research examining material hardship patterns and their association with both mental and physical health is urgently needed to comprehensively understand the well-being of older adults.

## Current study

Given the above issues, the aim of this paper is to expand our understanding of material hardship specifically by identifying the multidimensional occurrences of material hardship over time and examining its association with mental and physical health. In this research, we used the HRS to identify patterns of material hardship among older adults. Multi-channel sequence analysis (MCSA) is a powerful methodological approach for examining complex, cumulative effects across various life domains. For example, MCSA has been employed to classify temporary employment careers by integrating income and employment trajectories, revealing diverse career patterns in the Netherlands [8]. Additionally, MCSA has been utilized to examine the relationship between work-family life-course patterns and later-life employment, emphasizing the significance of continuous employment and the timing of family transitions [9].

We utilize MCSA to model simultaneous material hardship trajectories. This approach incorporates multiple distinct measures of material hardship, including whether older adults experienced a lack of food, cutting back on medications because of cost, difficulty paying bills, stress on the financial situation, and housing expenses. This approach is particularly helpful as it conceptualizes material hardship, focusing on trajectories with their condition changes within different material hardship domains. We also examine how material hardship trajectories affect the well-being of older adults by considering the persistence and timing of these hardships throughout their lives. Specifically, we are interested in (a) What types of material hardship trajectories emerged among older adults? (b) How are the types of material hardship trajectories associated with mental and physical health among older adults?

## Methods

### Data source

In this research, HRS data is used to examine the association between material hardship sequences and their association with mental health and physical health. HRS is a nationally representative longitudinal aging study that surveys more than 37,000 adults 50 years and older and their spouses/partners in the United States. HRS is sponsored by the National Institute on Aging and performed by the Institute for Social Research at the University of Michigan (U01 AG009740) [10]. The University of Michigan Institutional Review Board granted approval for the HRS study, ensuring adherence to the principles outlined in the Declaration of Helsinki. All participants provided informed consent prior to the interviews, in compliance with ethical approval and participation guidelines. The Leave Behind (LB) module was included, which investigates psychosocial information in each wave from a rotating 50% of the participants [11]. 3 waves of HRS data (2010, 2014, and 2018) were used due to the availability of LB module (The LB module data collects responses of the same individuals responding once every four years.). The research sample included those aged 65 and above in 2010 and completed all responses in 2010, 2014, and 2018. In addition, the sample was restricted to the financial respondents of the household in order to reduce the possible interdependence of each individual in estimating financial hardship. Financial respondents were in charge of answering the income-related questionnaires [10]. The final sample was composed of 2,628 individuals and multiple imputation was conducted to deal with missing cases (about 5% cases) with the final sample.

### Measurement

**Independent variable.** We use five material hardships directly available in the core HRS and LB module questions following concepts: (1) difficulty with bills, (2) food insecurity, (3) medication cutback experience, (4) financial strain and (5) unaffordable housing. To be specific, the first one refers to difficulty with bills and it asks the following question: "How difficult is it for (you/ your family's) to meet monthly payments on your (family's) bills?" and the five category answers were from not at all difficulty to completely difficult. The second one is about food security using a question. It asks the question, "In the last two years, have you always had enough money to buy the food you need?" with a yes-no answer. The third one asks about medication cutback experience with the question, "At any time in the last two years, have you ended up taking less medication than was prescribed for you because of the cost?" with a yes-no answer. The fourth one refers to financial strain with the question, "If an ongoing financial strain was a current and ongoing problem that has lasted twelve months or longer?" with four category answers from no to very upsetting. The last unaffordable housing was assessed if their rent or mortgage payments exceeded 30% of their gross household income [12].

**Dependent variable.** Two dependent variables in 2018 (last wave) were used in the study: depressive symptoms for mental health and self-rated health for physical health. The Depressive symptoms were measured with a subset of the CES-D. Using the 8-item version of the instrument, respondents were asked to respond yes/no as to whether (a) they were depressed, (b) everything "felt like an effort," (c) their sleep was restless, (d) they were happy, (e) they were lonely, (f) they enjoyed life, (g) they felt sad, and (h) they could not "get going." A summary score (range 0-8) was constructed with the reverse coding of positive items, and higher scores reflect more depressive symptoms.

Self-rated health was measured with a five-point scale from excellent (= 1) to very poor (= 5). A higher score represents worse self-rated health.

**Covariates.** We controlled for demographic characteristics that have been shown to be associated with material hardship. Marital status was categorized into two categories: married and others. Poverty status was categorized as poor (income-to-poverty ratio ≤ 1), near poor (income-to-poverty ratio 1–1.25). or not poor (income-to-poverty ratio > 1.25). Gender was assessed with a binary variable of male (0) or female (1). Education was coded into three categories: Less than high school, high school, and college and above. Age and the number of chronic diseases were used as a continuous variable (range 0–8). Race/ethnicity was coded as white, black, Hispanic, and others. Working status in 2018 was categorized into two categories: working and others.

## Statistical analyses

The analytic methodologies used to investigate the in-depth association between material hardship patterns, depressive symptoms, and self-rated health encompass three approaches: 1. multi-channel sequence analysis (MCSA), 2. bivariable analysis, and 3. ordinary least squares (OLS) regression analysis.

**1) Multi-channel sequences analysis (MCSA).** MCSA is used to identify multidimensional material hardship, creating sequence data for each individual. MCSA is a sequence analysis that deals with sequences in multiple parallel domains simultaneously [13] and is particularly helpful in understanding the dynamics of material hardship in later life. Sequence analysis has become increasingly important in the field of social sciences as a powerful tool by providing a visual representation of index plots that display the evolution of the cross-sectional distribution at successive time points and show the dynamics in population-level studies [14]. Particularly, sequence analysis highlights the diversity of life patterns in a way that cannot be achieved with traditional statistical modeling [15].

Material hardship patterns were constructed for each individual using 3 waves of data (2010, 2014, and 2018); these were then categorized using Dynamic Hamming (DH) distances [16]. DH distances are a type of metric that represents the 'cost' of converting one person's sequences to that of another. As a result, they quantify how similar or distinct individual biographies are [17]. When calculating distances across sequences, avoiding insertion and deletions guarantees that the timing of transitions between alternate states is preserved. It also eliminates the arbitrary nature of researcher-based cost assignment for insertions and deletions. The DH algorithm requires complete data on the sequence variables [18]. To retain as many cases as possible, we imputed any missing parts of a sequence. Values were filled in using multiple imputations implemented in Stata.

Drawing on the pairwise distance matrix, cluster analysis was conducted to classify individual sequences into homogeneous types. The Ward hierarchical clustering method was used to individual agglomerate sequences and create sequence groups of individual sequences. To determine the most discriminant number of sequence types, we used insights from hierarchical cluster analysis measures and from the visual inspection of the dendrogram with the need to obtain a substantively interpretable solution following common practice in cluster analysis [19].

**2) Bivariate analysis.** Bivariate analysis was used to investigate the correlations between the key variables. Bivariate analysis using chi-square and ANOVA were used to examine the association between material hardship sequences and sociodemographic characteristics.

**3) Ordinary least squares (OLS) regression analysis.** OLS regression analysis was conducted to estimate the relationship between material hardship sequences and depressive symptoms or self-rated health. The OLS regression model included the independent variable (material hardship sequences), the dependent variables (depressive symptoms and self-rated health), and covariates (marital status, poverty status, gender, education, age, number of chronic diseases, race/ethnicity, and working status). All analyses were performed separately using Stata 14.0.

## Results

The descriptive statistics of the study's sample are presented in Table 1. Among the total sample, men accounted for 39.95% and women 60.05%. By race, most of the sample was White (75.27%), and by education, more than half of the sample had a high school diploma. Approximately 65% of the sample was married, and 24.7% were living alone. 10% of the sample was in poverty in 2010. The average age of the sample was 73.11.

### Sequence analysis

Five types of possible material hardship sequences were identified (Fig 1). The first type, *Least materially burdened* (n = 1,054, 40.11%), describes individuals who had no difficulty or very low difficulty (less than 3%) across domains over 3 waves. The second type, *Multiply burdened* (n = 323, 12.29%), depicts individuals who experienced persistently severe material hardship in multiple domains (bills, food, medication, and financial strain). The third type, *Financially burdened* (n = 486, 18.49%), refers to individuals who experienced about 20% of financial strain, while other domain experiences were lower than 10%. The fourth type, *Housing cost burdened* (n = 548, 20.85%), refers to individuals who experienced about 63% of unaffordable housing difficulty, while other difficult domains were lower than 5%. consistently experienced housing cost burden while other material hardship domains remained low at most time points. The last type, *Financial & housing cost burdened* (n = 217, 8.26%), pertains to individuals who experienced about 24% of ongoing financial strain and 85% of unaffordable housing, while other difficult domains were lower than 10%. This group of people faced severe material hardship condition in 2014 and showed a slight recovery in 2018.

**Table 1. Descriptive analysis.**

| | | Frequency | % |
|---|---|---|---|
| Gender | Male | 1,050 | 39.95 |
| | Female | 1,578 | 60.05 |
| Race/ethnicity | White | 1,978 | 75.27 |
| | Black | 361 | 13.74 |
| | Hispanic | 229 | 8.71 |
| | others | 60 | 2.28 |
| Education | Below Highschool | 485 | 18.46 |
| | Highschool | 982 | 37.38 |
| | Above Highschool | 1,160 | 44.16 |
| Marital status (2010) | Married | 1,699 | 64.65 |
| | other | 929 | 35.35 |
| Living alone (2010) | Living alone | 649 | 24.70 |
| | other | 1,979 | 75.30 |
| Poverty (2010) | Poor | 199 | 7.57 |
| | Near poor | 89 | 3.39 |
| | Not poor | 2,340 | 89.04 |
| Work status (2018) | Working | 247 | 9.41 |
| | Not working | 2,378 | 90.59 |
| | | Average | SD |
| Number of chronic disease (2010) | | 2.19 | 1.32 |
| Age (2010) | | 73.11 | 5.70 |
| Self-rated health (2018) | | 3.04 | 1.02 |
| Depressive symptoms (2018) | | 1.32 | 1.84 |

## Group 1: Least materially burdened

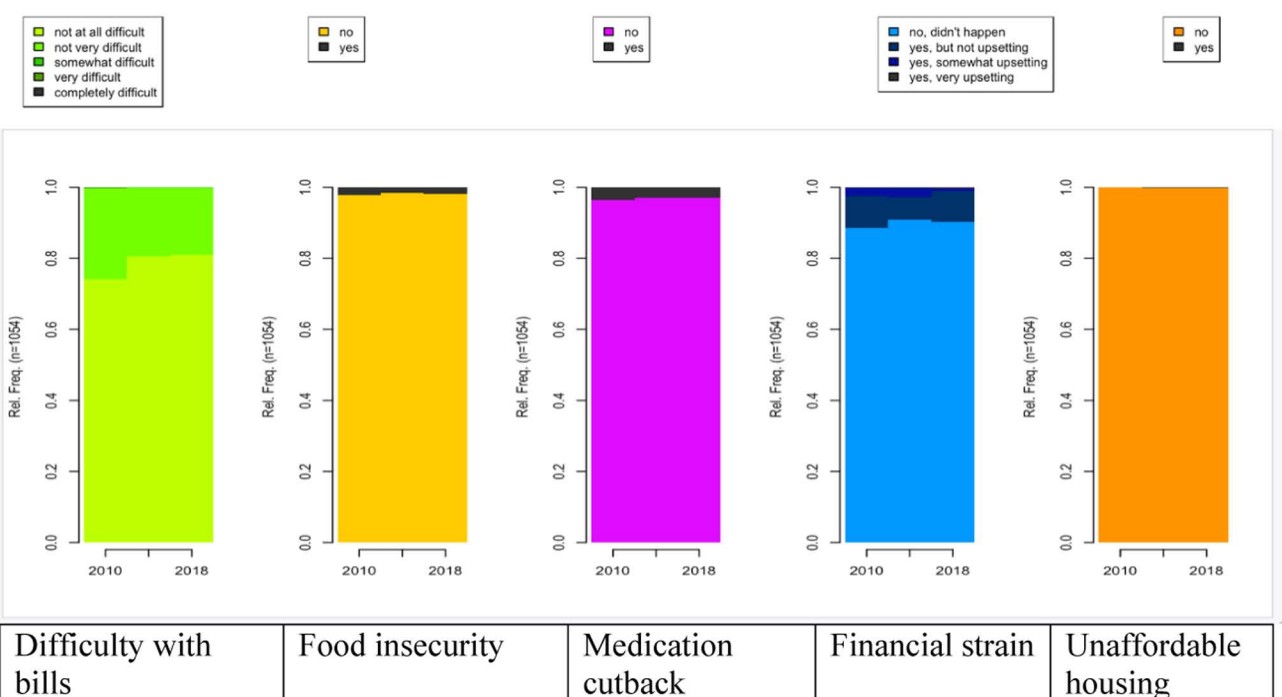

| Difficulty with bills | Food insecurity | Medication cutback | Financial strain | Unaffordable housing |

## Group 2: Multiply burdened

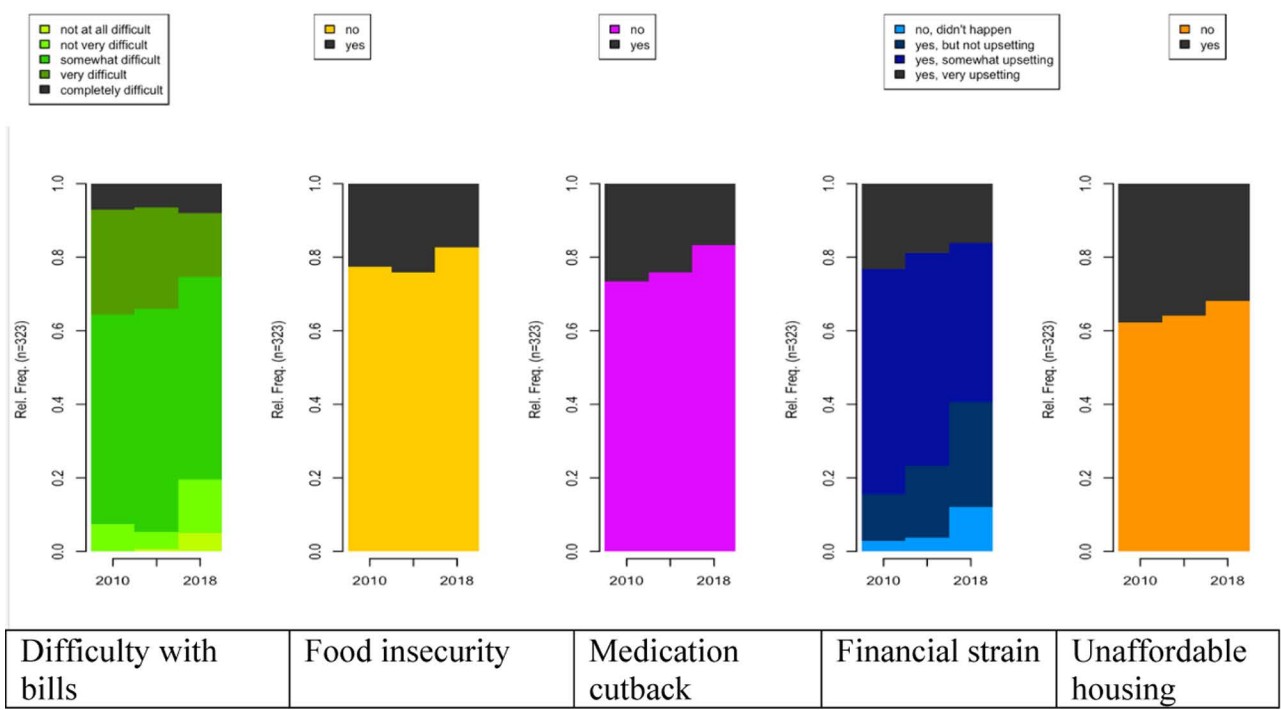

| Difficulty with bills | Food insecurity | Medication cutback | Financial strain | Unaffordable housing |

## Group 3: Financially burdened

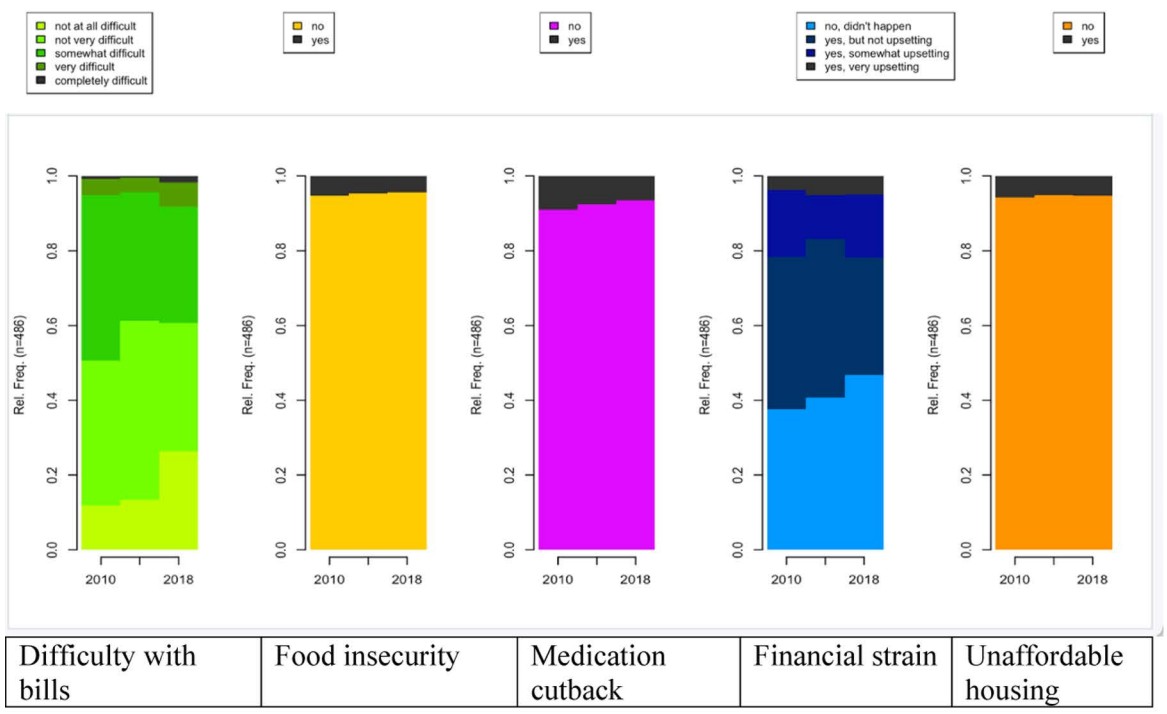

| Difficulty with bills | Food insecurity | Medication cutback | Financial strain | Unaffordable housing |

## Group 4: Housing cost burdened

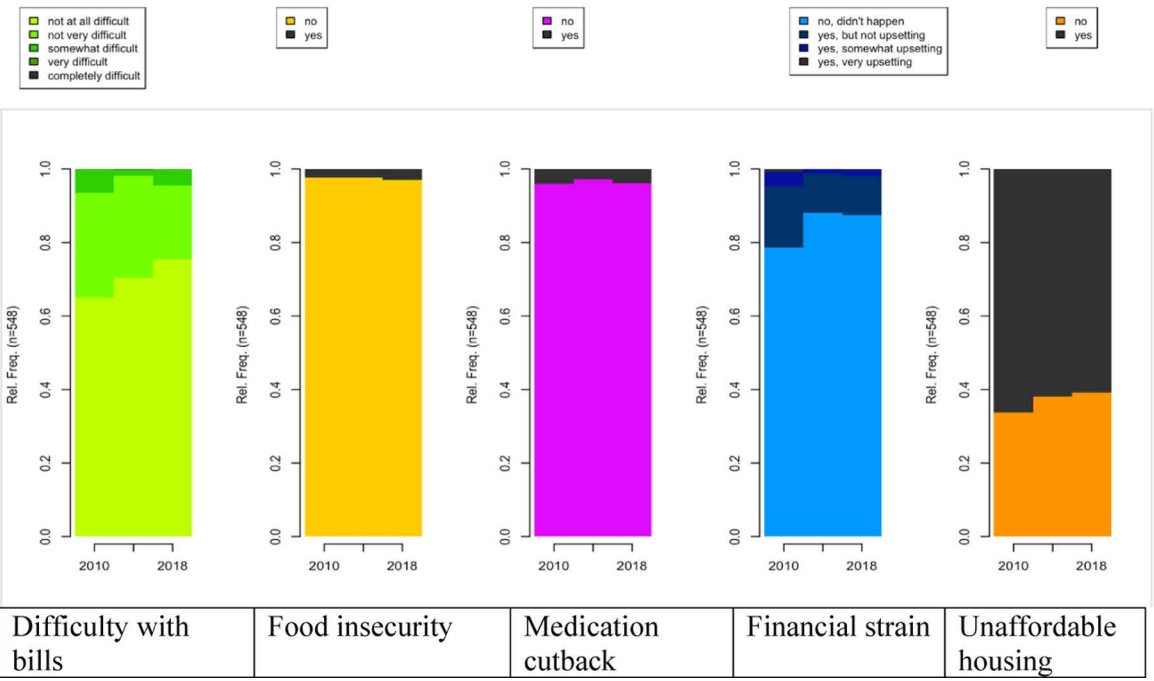

| Difficulty with bills | Food insecurity | Medication cutback | Financial strain | Unaffordable housing |

Group 5: Financial & housing cost burdened

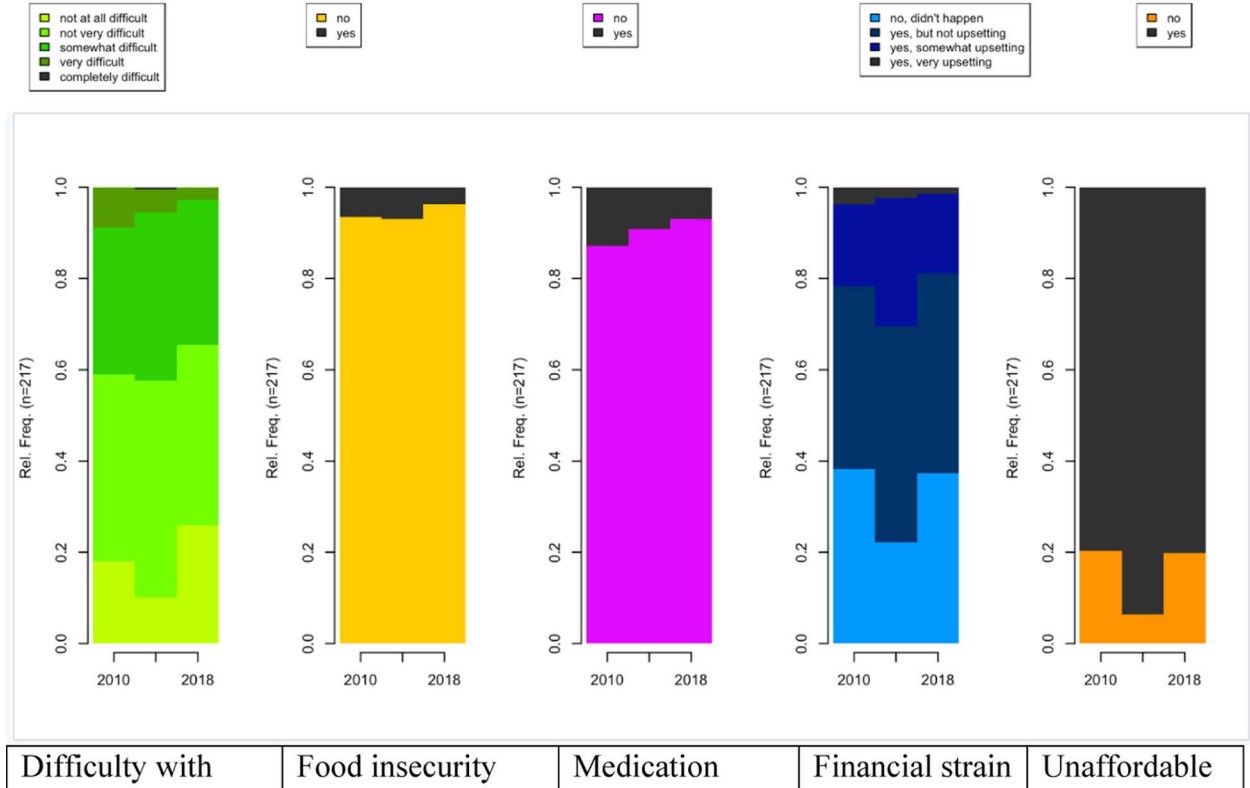

**Fig 1. Sequence analysis results.**

## Bivariate analysis

Table 2 shows the bivariate analysis results. *Least materially burdened* was associated with being White (83.40%), married (70.59%), not poor (92.88%), and least number of chronic diseases (2). *Multiply burdened* was related to being female (65.63%), having below high school education (25.39%), not married (44.89%), living alone (27.86%), poor (13.93%), and having the most chronic diseases (2.58). This group of people was the most vulnerable population in terms of socioeconomic status. *Financially burdened* was associated with being female (64.81%), below high school (25.36%), living alone (27.57%), poor (10.91%), and near poor (6.58%).

*Housing cost-burdened* was associated with having an education above high school (53.65%). Although the people in this group have experienced hardship, they are not as disadvantaged based on bivariate analysis. The last group, *Financial & housing cost burdened* was associated with being male (42.86%), Black (25.35%), working (15.67%), and living alone (27.19%).

## Ordinary least squares (OLS) regression analysis

**Depressive symptoms.** Table 3 shows the regression analysis results with depressive symptoms. The *Multiply burdened* (b = 0.68, p ≤ 0.001), *Financially burdened* (b = 0.28,

**Table 2. Bivariate analysis results.**

| | | Least materially burdened (40.11%) | Multiply burdened (12.29%) | Financially burdened (18.49%) | Housing cost-burdened (20.85%) | Financial & housing cost-burdened (8.26%) | Sig (chi-square/ ANOVA test) |
|---|---|---|---|---|---|---|---|
| Gender | Male | 42.60 | 34.37 | 35.19 | 41.24 | 42.86 | 13.03* |
| | Female | 57.40 | 65.63 | 64.81 | 58.76 | 57.14 | |
| Race/ethnicity | White | 83.40 | 63.16 | 70.16 | 76.28 | 62.67 | 101.42*** |
| | Black | 9.11 | 20.74 | 16.26 | 11.68 | 25.35 | |
| | Hispanic | 5.79 | 12.69 | 10.08 | 10.22 | 10.14 | |
| | others | 1.71 | 3.41 | 3.50 | 1.82 | 1.84 | |
| Education | Below HS | 15.09 | 25.39 | 25.36 | 15.33 | 17.05 | 73.53*** |
| | High School | 38.52 | 40.56 | 41.24 | 31.02 | 34.56 | |
| | Above HS | 46.39 | 34.06 | 33.40 | 53.65 | 48.39 | |
| Work status (2010) | Working | 7.69 | 10.84 | 8.66 | 10.05 | 15.67 | 14.97** |
| | Not working | 92.31 | 89.16 | 91.34 | 89.95 | 84.33 | |
| Marital status (2010) | Married | 70.59 | 55.11 | 60.29 | 64.78 | 59.45 | 35.75*** |
| | other | 29.41 | 44.89 | 39.71 | 35.22 | 40.55 | |
| Living alone (2010) | Living alone | 22.01 | 27.86 | 27.57 | 24.45 | 27.19 | 8.73 |
| | Other | 77.99 | 72.14 | 72.43 | 75.55 | 72.81 | |
| Poverty (2010) | Poor | 5.22 | 13.93 | 10.91 | 5.66 | 6.91 | 66.58*** |
| | Near poor | 1.9 | 3.41 | 6.58 | 2.55 | 5.53 | |
| | Not poor | 92.88 | 83.66 | 82.51 | 91.79 | 87.56 | |
| Number of chronic disease (2010) | | 2 (1.23) | 2.58 (1.4) | 2.4 (1.39) | 2.1 (1.27) | 2.35 (1.37) | 17.38*** |
| Age (2010) | | 73.71 (5.65) | 72.64 (5.25) | 73 (5.49) | 73.17 (6.13) | 70.92 (5.35) | 11.69*** |
| Depressive symptoms (2018) | | 1.04 (1.62) | 1.99 (2.18) | 1.51 (1.90) | 1.22 (1.80) | 1.55 (1.96) | 19.82*** |
| Self-rated health (2018) | | 2.88 (0.99) | 3.43 (1.08) | 3.22 (1.0) | 2.93 (0.99) | 3.01 (0.99) | 23.87*** |

*** p ≤0.001; ** p ≤0.01; * p ≤ 0.05.

p ≤ 0.01), *Financial & housing cost burdened* (b = 0.44, p ≤ 0.001) were associated with a higher level of depressive symptoms compared to the *Least materially burdened.* Furthermore, being female (b = 0.24, p ≤ 0.01), being Hispanic (b = 0.53, p ≤ 0.01), being poor (b = 0.35, p ≤ 0.05), being older (b = 0.02, p ≤ 0.05), and having more chronic disease (b = 0.24, p ≤ 0.001) were associated with a high level of depressive symptoms while having a higher education was associated with a lower level of depressive symptoms (b = -0.34, p ≤ 0.001; b = -0.47, p ≤ 0.001).

We conducted an additional analysis of different reference groups (S1 Table) to further investigate the association between material hardship and mental health. We found the deleterious effect of multiple material hardships on depressive symptoms among older adults. *Least materially burdened* (b = -0.68, p ≤ 0.001), *Financially burdened* (b = -0.4, p ≤ 0.01), and *Housing cost burdened* (b = -0.53, p ≤ 0.001) were associated with the lower level of depressive symptoms compared to *Multiply burdened*. In addition, *Multiply burdened* (b = 0.53, p ≤ 0.001) and *Financial & housing cost-burdened* (b = 0.3, p ≤ 0.05) were associated with a higher level of depressive symptoms compared to the *Housing cost burdened*.

**Self-rated health.** Table 4 shows the regression analysis results with self-rated health. In terms of self-rated health, the *Multiply burdened* (b = 0.32, p ≤ 0.001) and *Financially burdened* (b = 0.181, p ≤ 0.001) were associated with worse self-rated health compared to the *Least materially burdened*. Furthermore, being Black (b = 0.20, p ≤ 0.001), being Hispanic (b = 0.26, p ≤ 0.001), being poor (b = 0.2, p ≤ 0.001), being older (b = 0.01, p ≤ 0.01), and having more

**Table 3. Regression analysis results with depressive symptoms.**

|  | Coefficient | S.E | 95% Cl |
|---|---|---|---|
| *Material hardship sequences (ref: Least materially burdened)* | | | |
| Multiply burdened | 0.68*** | 0.11 | 0.45 – 0.9 |
| Financially burdened | 0.28** | 0.1 | 0.09 – 0.47 |
| Housing cost burdened | 0.14 | 0.09 | −0.04 – 0.33 |
| Financial & housing cost burdened | 0.44*** | 0.13 | 0.18 – 0.7 |
| *Gender (ref: male)* | | | |
| *Female* | 0.24*** | 0.07 | 0.1 – 0.38 |
| *Marital status (ref: others)* | | | |
| Married | −0.04 | 0.12 | −.027 – 0.19 |
| Living alone *(ref: others)* | | | |
| Living alone | −0.18 | 0.12 | −0.42 – 0.06 |
| *Race/ethnicity (ref: White)* | | | |
| Black | 0.06 | 0.11 | −0.15 – 0.27 |
| Hispanic | 0.53*** | 0.13 | 0.26 – 0.79 |
| Others | 0.03 | 0.23 | −0.43 – 0.48 |
| *Education (ref: below high school)* | | | |
| High school | −0.34*** | 0.1 | −0.55 – −0.14 |
| Above high school | −0.47*** | 0.11 | −0.67 2013 −0.26 |
| *Poverty (ref: non poor)* | | | |
| Near poor | 0.11 | 0.2 | −0.27 – 0.5 |
| Poor | 0.35* | 0.14 | 0.07 – 0.62 |
| *Working status (ref: others)* | | | |
| Working | −0.2 | 0.12 | −0.44 – 0.03 |
| Number of chronic disease | 0.24*** | 0.02 | 0.19 – 0.29 |
| Age | 0.02* | 0.01 | 0.004 – 0.03 |
| Constant | −0.55 | | |

*** p ≤0.001; ** p ≤0.01; * p ≤ 0.05.

chronic disease (b = 0.24, p ≤ 0.001) were associated with worse self-rated health while having above high school education (b = -0.29, p ≤ 0.001), and working (b = -0.32, p ≤ 0.001) were associated with better self-rated health.

With our additional analysis using different reference groups (S2 Table), we found out the negative effect of multiple material hardships on self-rated health among older adults. *Least materially burdened* (b = -0.32, p ≤ 0.001), *Financially burdened* (b = -0.14, p ≤ 0.05), *Housing cost burdened* (b = -0.31, p ≤ 0.001), and *Financial & housing cost burdened* (b = -0.23, p ≤ 0.01) were associated with better self-rated health compared to the *Multiply burdened*. In addition, *Multiply burdened* (b = 0.31, p ≤ 0.01) and *Financially burdened* (b = 0.17, p ≤ 0.001) were associated with poor self-rated health compared to the *Housing cost-burdened*.

## Discussion

Many existing studies explore the relationship between poverty and health, emphasizing the long-term effects of various poverty pathways on health. Assessing financial situations based on poverty status may not fully capture the unique circumstances of older adults. To address the existing gap in the literature, we examined the material hardship trajectories and their relationship with two health indicators (mental and physical health) to provide

**Table 4.  Regression analysis results with self-rated health.**

|  | Coefficient | S.E | 95% Cl |
|---|---|---|---|
| *Material hardship sequences (ref: Least materially burdened)* | | | |
| Multiply burdened | 0.31*** | .06 | 0.2 – 0.43 |
| Financially burdened | 0.18*** | 0.05 | 0.08 – 0.28 |
| Housing cost burdened | 0.01 | 0.05 | −0.08 – 0.1 |
| Financial & housing cost burdened | 0.08 | 0.07 | −0.05 – 0.22 |
| *Gender (ref: male)* | | | |
| *Female* | −0.004 | 0.04 | −0.08 – 0.07 |
| *Marital status (ref: others)* | | | |
| Married | 0.0005 | 0.06 | −0.12 – 0.12 |
| *Living alone (ref: others)* | | | |
| Living alone | −0.07 | 0.06 | −0.2 – 0.05 |
| *Race/ethnicity (ref: White)* | | | |
| Black | 0.2*** | 0.06 | 0.09 – 0.31 |
| Hispanic | 0.26*** | 0.07 | 0.12 – 0.39 |
| Others | 0.19 | 0.12 | −0.04 – 0.43 |
| *Education (ref: below high school)* | | | |
| High school | −0.16** | 0.05 | −0.27 – −0.06 |
| Above high school | −0.29*** | 0.05 | −0.39 – −0.18 |
| *Poverty (ref: non poor)* | | | |
| Near poor | 0.16 | 0.1 | −0.04 – 0.36 |
| Poor | 0.2** | 0.07 | 0.06 – 0.35 |
| *Working status (ref: others)* | | | |
| Working | −0.32*** | 0.06 | −0.43 – −0.19 |
| Number of chronic disease | 0.24*** | 0.01 | 0.22 – 0.27 |
| Age | 0.01** | 0.003 | 0.002 – 0.02 |
| Constant | 1.77 | | |

*** $p \leq 0.001$; ** $p \leq 0.01$; * $p \leq 0.05$.

comprehensive information about a person's material hardship experience. The experience of material disadvantage can be made more evident by identifying "patterns of material hardship" with a long-term perspective, which provides evidence of the cumulative consequences of material disadvantage over time.

As the first examination of longitudinal patterns of material hardship among older adults using multidimensional domains, this study leveraged 3 waves of data to identify new and conceptually meaningful patterns of material hardship. The five patterns that emerged describe older adults' experiences of stability in varying intensities of hardship and of change over time in hardship intensity. Among the five types, the group that experienced the most difficulties in multiple domains was *Multiply burdened,* and the characteristics of this group were associated with females, low education, living alone, and poverty. It is noteworthy that *Multiply burdened*, *Financially burdened*, and *Financial & housing cost burdened* groups have experienced 'change' during the observation period. Specifically, the *Multiply burdened* and *Financially burdened* groups showed slight improvement, whereas *the Financial & housing cost burdened* group experienced inconsistent changes, indicating different trajectories among the groups. The change patterns of those two groups seem to be associated with sociodemographic factors. Comparing *Multiply burdened* group, composed of less educated, poor, and

unhealthy females, with the *Financial & housing cost burdened* group, consisting of working Black males, suggests that these dynamics may stem from an interplay with gender, education, and racial/ethnicity.

In addition, two different types of housing cost burden related groups (*Housing cost burdened, Financial & housing cost burdened*) were important findings of the research that capture the unique characteristics of the material hardship experience among older adults. The individuals in this group seem to address basic needs such as food and medical care; however, they are experiencing hardship related to housing and financial strain, which may reflect the high cost of living or expensive housing prices. Interestingly the *Housing cost burdened* group had higher education levels and a higher portion of White individuals who were not working, while the *Financial & housing cost burdened* had a higher proportion of Black individuals and higher education with working conditions, which implies that Black individuals experienced more severe financial stress related to housing costs. Especially, *Financial & housing cost burdened* group showed that their experience with financial strain and unaffordable housing became severe, especially in 2014. A possible explanation of the phenomenon can be the dramatic increase in housing prices in 2014, which makes it difficult for older adults on fixed incomes to afford housing [20]. Older homeowners who fall behind on their mortgage payments are at risk of foreclosure. In recent years, homeowners aged 75 and older have had the highest foreclosure rates [21], and especially older Black and Hispanic homeowners are considered at a high risk of paying off their mortgages in their old ages [22] which supports the finding of the study. The findings of this study provide a novel and multifaceted conceptualization of deprivation and valuable evidence about the differential impacts of social demographic characteristics on older adults' experiences of material hardship.

Furthermore, the results revealed innovative findings regarding the connection between these patterns and both mental and physical health, offering a comprehensive understanding of well-being among older adults. We identified that patterns inimical to mental and physical health are most prevalent among those in *Multiply burdened*, and *Financially burdened* groups, which implies the detrimental effect of the cumulate material hardship experiences. The result emphasizes the need for further investigations of individuals in these groups. Especially, *Multiply burdened* possibly leads to poor mental/physical health amidst multiple hardships, attributed to insufficient adoption of a suitable healthy lifestyle across various domains and limited access to healthcare [23]. In addition, identifying the negative association between the *Financially burdened* group and mental/physical health implies that, beyond objective poverty measures, there is significance in acknowledging the subjective constraints older adults face concerning cash flow and financial stress. This result is supported by the existing literature on the relationship between long-term material stressors and a number of health problems [3,24,25]. Especially, the results underscore the significance of the duration and persistence of material hardship on well-being. The findings suggest that prolonged stress, possibly combined with limited existing resources, could have an adverse impact on overall health and well-being.

In addition, the finding that the *Financial & housing cost burdened* group was only associated with mental health, and not with physical health, provides the importance of the negative impact of financial stress related to housing costs on mental health. This refers to the importance of understanding recurring material related stressors, especially financial strain and housing costs. The results are supported by existing literature that shows persistent housing cost burdens and experiences are associated with increased risk of health among older adults [26]. Marginalized in policy and support strategies, possibly due to their higher income level and employment status, it could greatly benefit from further research tailored to meet the needs of older adults within it.

The findings presented in this paper need to be interpreted with some limitations. Due to data availability, we only used three waves over an 8-year period. In the future, considering more information over a longer term can provide a better understanding of material hardship trajectories and their impact on the physical and mental health of older adults.

Additionally, our results may oversimplify the association between material hardships and their impact on health, potentially excluding some confounding factors. There is a high probability that cumulative effects stem not only from material hardship experiences but also from associated areas of life, such as problems in family and social relationships and self-esteem related to material hardship [3]. Future research should aim to identify mechanisms in these associations using potential mediators, such as psychosocial factors, including self-efficacy (social, financial, health). This study investigates the relationship between patterns of material hardship and later-life well-being, emphasizing the need for future research to examine cumulative effects for a more comprehensive understanding.

## Conclusion

Despite these limitations, this is the first study we know to uncover the long-term trajectories of material hardship and their association with well-being experienced by older adults. By examining corresponding socio-demographic characteristics, the research identifies groups experiencing chronic difficulties, with a particular emphasis on highlighting the severity of housing-related issues. According to the U.S. Census Bureau, in 2022, the national poverty rate for older adults was 10.3% [27]. This indicates that *Multiply burdened* may fall into the low-income category, making them eligible for assistance programs. However, there is a significant need for future policies supporting *Financially burdened* and *Financial & housing cost burdened*, as these groups may face material hardship in their later life and may not be eligible for government support systems.

Additionally, by empirically identifying how stress related to material hardship, especially housing-related issues, adversely affects the psychological and physical health of groups experiencing long-term difficulties, this study provides valuable information for future implications and policies. For example, in addressing material hardship among older adults, developing comprehensive assessment tools should consider not only traditional indicators such as meals and bills but also factors like current mortgage-related challenges in the context of rapidly increasing housing prices and rents. Moreover, efforts toward the development of programs aimed at promoting the mental and physical well-being of older adults may be needed, addressing chronic material hardship and housing-related issues simultaneously in the future.

## Supporting information

**S1 Table. Regression analysis results with depressive symptoms.**
(DOCX)

**S2 Table. Regression analysis results with self-rated health.**
(DOCX)

## Author contributions

**Conceptualization:** Oejin Shin, Eunsun Kwon, Seoyeon Ahn, Sojung Park.

**Data curation:** Oejin Shin.

**Formal analysis:** Oejin Shin.

**Methodology:** Oejin Shin, Eunsun Kwon, Seoyeon Ahn.

**Supervision:** Eunsun Kwon, Seoyeon Ahn, Sojung Park.

**Writing – original draft:** Oejin Shin.

**Writing – review & editing:** Oejin Shin, Eunsun Kwon, Seoyeon Ahn, Sojung Park.

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
