## [Decision Letter · Decision Letter 0]

17 Oct 2024

PONE-D-24-42119The Association Between Material Hardship and Physical and Mental Health among Older Adults: Multi-Channel Sequence ApproachPLOS ONE

Dear Dr. Shin,

Thank you for submitting your manuscript to PLOS ONE. After careful consideration, we feel that it has merit but does not fully meet PLOS ONE’s publication criteria as it currently stands. Therefore, we invite you to submit a revised version of the manuscript that addresses the points raised during the review process.

We look forward to receiving your revised manuscript.

Kind regards,

Binh Thang Tran, MPH, PHD

Academic Editor

PLOS ONE

2. Please amend your manuscript to include your abstract after the title page.

Reviewers' comments:

Reviewer's Responses to Questions

**Comments to the Author**

1. Is the manuscript technically sound, and do the data support the conclusions?

Reviewer #1: Yes

Reviewer #2: Yes

2. Has the statistical analysis been performed appropriately and rigorously? 

Reviewer #1: Yes

Reviewer #2: Yes

3. Have the authors made all data underlying the findings in their manuscript fully available?

Reviewer #1: Yes

Reviewer #2: Yes

4. Is the manuscript presented in an intelligible fashion and written in standard English?

Reviewer #1: Yes

Reviewer #2: Yes

5. Review Comments to the Author

Reviewer #1: It’s an interesting and innovative study that identifies material hardship over time and examines its association with mental and physical health. Nevertheless, some minor points are unclear and, in some cases, need to be revised:

I recommend this manuscript for minor revision.

1. The relationship between material hardship and both mental and physical health can be considered in different contexts and over time. The current study uses available data from the HRS and LB modules; however, the outcomes and factors of interest may differ across time periods. Because material disadvantages accumulate and fluctuate over time, the authors should discuss and explain these cumulative consequences more clearly for the readers to understand

2. How potential confounding factors are adjusted in the Ordinary Least Squares (OLS) regression analysis should be described more clearly

Reviewer #2: First impressions

Overall, the research design is sound and the writing is well-structured. The background section is excessively lengthy and requires condensation. The study provides interesting information about material hardship and its association with the well-being experienced by older adults. The paper was well structured with the appropriate language used. The study results addressed the research question and objective.

However, there are some specific issues within each section as detailed in the comments below.

Introduction: Material hardship among older adults

The introduction was well-structured and provides comprehensive information regarding the paper, as well as was well organized and explained the importance of the study. Some points should be considered to improve the introduction:

Line 2 of Page 3: Authors should cite references.

Line 3 of Page 4: The author should add a few studies that have used multichannel chain analysis to simultaneously model complicated material trajectories, thereby demonstrating this method's usefulness.

Methodology

- Abbreviations: abbreviations noted the first time; From the second time onwards, only write that symbol: Health and Retirement Study (HRS); multi-channel sequence analysis (MCSA); ordinary least squares (OLS);....

- Statistical analyses:

+ Descriptive statistics: Applied to describe data about research subjects' information according to fall risk status: Number, rate (%); ± SD for variables following normal distribution; Median (range) for variables that do not follow a normal distribution.

+ Inferential statistics: statistically significant with: *** p<0.001; ** p <0.01; * p < 0.05 . Univariate and multivariate regression analysis to learn some factors related to the risk of falls in the elderly (statistically significant with 95%CI and *** p<0.001; ** p <0.01; * p < 0.05).

Results and discussion

- In Table 1:

+ Education: Below high school (485) + High school (982) + Above High school (1160) ≠ 2628 participants

+ Work status (2018): Working (247) + Not working (2378) ≠ 2628 participants

- In Table 3, the authors should add a footnote of the table: chi-square and ANOVA test, *** p<0.001; ** p <0.01; * p < 0.05.

Conclusion

Acceptable.

References, tables and figures

In the reference “Halpin, B. (2019)” shouldn’t delete DOI. Many references are outdated and need to be replaced by newer references.

Authors should conform to the format of the References as required by the Journal.

6. PLOS authors have the option to publish the peer review history of their article (what does this mean? ). If published, this will include your full peer review and any attached files.

**Do you want your identity to be public for this peer review?** For information about this choice, including consent withdrawal, please see our Privacy Policy .

Reviewer #1: No

Reviewer #2: No

---

## [Author Response · Author response to Decision Letter 0]

18 Dec 2024

We greatly appreciate the opportunity to revise the above-referenced manuscript. We have reviewed the comments thoroughly and used them as a basis for our revisions. With these revisions, we hope that the manuscript may be considered for publication in PLOS ONE.

Please note that our responses are blue in standard format if they are presented in the letter only but italicized if they are included in the manuscript. In the revised manuscript, we highlighted in yellow any text that has been revised or added in the response letter word file.

---

## [Decision Letter · Decision Letter 1]

30 Jan 2025

The Association Between Material Hardship and Physical and Mental Health among Older Adults: Multi-Channel Sequence Approach

PONE-D-24-42119R1

Dear Dr. Shin,

We’re pleased to inform you that your manuscript has been judged scientifically suitable for publication and will be formally accepted for publication once it meets all outstanding technical requirements.

Kind regards,

Associate Professor Dr. Nik Ahmad Sufian Burhan

Academic Editor

PLOS ONE

Additional Editor Comments (optional):

Reviewers' comments:

Reviewer's Responses to Questions

**Comments to the Author**

1. If the authors have adequately addressed your comments raised in a previous round of review and you feel that this manuscript is now acceptable for publication, you may indicate that here to bypass the “Comments to the Author” section, enter your conflict of interest statement in the “Confidential to Editor” section, and submit your "Accept" recommendation.

Reviewer #2: All comments have been addressed

Reviewer #3: All comments have been addressed

2. Is the manuscript technically sound, and do the data support the conclusions?

Reviewer #2: Yes

Reviewer #3: Yes

3. Has the statistical analysis been performed appropriately and rigorously? 

Reviewer #2: Yes

Reviewer #3: Yes

4. Have the authors made all data underlying the findings in their manuscript fully available?

Reviewer #2: Yes

Reviewer #3: Yes

5. Is the manuscript presented in an intelligible fashion and written in standard English?

Reviewer #2: Yes

Reviewer #3: Yes

6. Review Comments to the Author

Reviewer #2: The reviewer considers that the manuscript was improved.

I hope that the manuscript may be considered for publication in PLOS ONE.

Reviewer #3: I have reviewed both the original and revised (R1) manuscripts. In general, I share the concerns raised by the previous reviewers. However, the authors have adequately addressed all of these comments in the revised manuscript. I find the manuscript to be of sound quality and recommend it to be accepted for publication.

7. PLOS authors have the option to publish the peer review history of their article (what does this mean? ). If published, this will include your full peer review and any attached files.

**Do you want your identity to be public for this peer review?** For information about this choice, including consent withdrawal, please see our Privacy Policy .

Reviewer #2: No

Reviewer #3: No

---

## [Editor Report · Acceptance letter]

PONE-D-24-42119R1

PLOS ONE

Dear Dr. Shin,

I'm pleased to inform you that your manuscript has been deemed suitable for publication in PLOS ONE. Congratulations! Your manuscript is now being handed over to our production team.

Kind regards,

on behalf of

Dr. Nik Ahmad Sufian Burhan

Academic Editor

PLOS ONE